# Longitudinal Process of Setting and Achieving Activity- and Participation-Level Goals in Home Rehabilitation in Japan: A Qualitative Study Using Trajectory Equifinality Modeling

**DOI:** 10.3390/ijerph20095746

**Published:** 2023-05-08

**Authors:** Yuki Saito, Kounosuke Tomori, Tatsunori Sawada, Kanta Ohno

**Affiliations:** 1Department of Rehabilitation Science, Division of Occupational Therapy, Sendai Seiyogakuin College, Miyagi 984-0022, Japan; 2Department of Occupational Therapy, School of Health Science, Tokyo University of Technology, Tokyo 152-8550, Japan; tomoriks@stf.teu.ac.jp (K.T.); sawadatn@stf.teu.ac.jp (T.S.); ohnoknt@stf.teu.ac.jp (K.O.)

**Keywords:** home rehabilitation, trajectory equifinality modeling, goal setting, goal attainment

## Abstract

This study aimed to clarify the longitudinal goal negotiation and collaboration process of achieving activity- and participation-level goals. We conducted a qualitative study using the trajectory equifinality model. Nine occupational therapists with experience in setting and achieving activity- and participation-level goals were recruited and interviewed about their clients. We identified two phases and four pathways in the setting and attainment process for activity- and participation-level goals. Throughout the longitudinal goal-setting process, when the occupational therapist and client had difficulty discussing activity- and participation-level goals, the therapist respected the client’s expectations, explained the purpose of occupational therapy in detail, and conducted individual face-to-face interviews. When it was difficult to provide work-based interventions, the occupational therapist made flexible use of functional training, elemental movement training, occupation-based practice, and environmental modifications. The results of this study may assist in supporting clients to improve their activity and participation in home rehabilitation.

## 1. Introduction

The traditional approach to goal setting in healthcare relies on a biomedical model wherein clinicians make decisions independently, leading to conflicting expectations between patients and healthcare providers, and frequently resulting in patients’ goals being unfulfilled. However, the current trend is towards a patient-centered approach to goal setting, which prioritizes patients’ needs and expectations. This collaborative process involves healthcare professionals, patients, and their families or caregivers as appropriate. Research has demonstrated that involving patients in goal formulation results in the development of more personalized goals and increased satisfaction with the rehabilitation experience [1].

According to Japan’s Ministry of Health, Labor, and Welfare [2], uniform functional training is provided as the primary objective in home rehabilitation settings, despite the diverse needs of clients, which may include functional recovery, engagement in occupation, and social participation. Therefore, individualized goal setting for activities and participation is recommended as a solution.

However, several studies conducted in the real world have revealed that goal setting is more complex and uncertain. Maitra and Erway’s study revealed all 11 occupational therapists aimed to involve their clients in goal setting; however, 14 (46.7%) of the 30 clients felt they had been involved in selecting less than 20% of their goals, and 13 (43.3%) of the 30 could not name all of their goals [3]. Similarly, Brown et al. reported that clients tended to have vague or broadly worded goals, and their uncertainty (about their condition and potential for recovery) frequently leads to challenges when setting goals [4]. Moreover, clients tend to set impairment-level goals more than activity or participation-level goals [1]. Thus, health professionals should continue to negotiate and collaborate with their clients to achieve their goals.

Ishikawa et al. [5] reported that daily activity experiences through assessment or intervention could help clients who could not express their occupational needs in the initial assessment to do so later. This finding suggests that recalling daily activity experiences facilitates their expectations for achieving activity- and participation-level goals. However, previous research on goal setting is mostly cross-sectional [1,3]; therefore, the longitudinal process for goal negotiation with the client remains unclear. This study aimed to clarify the longitudinal goal negotiation and collaboration process to achieve activity- and participation-level goals. The goals were as follows:

Clarify the process of occupational therapists’ assessment and intervention in achieving activity- and participation-level goals during home-based rehabilitation.Categorize the assessment and intervention processes occupational therapists utilize to achieve activity- and participation-level goals in home-based rehabilitation.

## 2. Materials and Methods

### 2.1. Study Design

We conducted a qualitative study using the trajectory equifinality model (TEM) developed by a Japanese researcher, Sato et al. [6,7,8], in the 2000s. TEM is a specialized qualitative methodology that captures human development in longitudinal (temporal), cultural, and social contexts, and presents it graphically as a tree diagram (Figure 1).

The TEM methodology requires that all participants reach the “equifinality point” (EFP), which we defined in this study as “the experience of attaining the activity/participation-level goal through occupational therapy.” Additionally, to identify participants who had experienced EFP, we used the historically structured inviting approach and interviewed them about their experiences in home rehabilitation. We employed TEM-specific conceptual terms (Table 1), including obligatory passage points (OPPs), bifurcation points (BFPs) at which the experience pathway splits into multiple branches, social guidance (SGs) that facilitates the forces of EFP, and social direction (SD) that inhibits the forces of EFP. We reported our findings following the Consolidated Criteria for Reporting Qualitative Research checklist [9].

### 2.2. Study Participants and Ethics

Researchers who developed TEM recommended the ideal number of participants, depending on the purpose of the study, using the “1, 4, 9 rule”, which indicates the following: one participant reveals the depth of individual experience; 4 ± 1 participants visualize the sprouts of diversity and reveal the diverse and convergent parts of the experience; and 9 ± 2 participants make the diversification of experience more complex, thereby making it suitable for typification [9].

We invited nine appropriate occupational therapists following the following criteria: (1) actively engaging in practices that focused on activities and participation in a home rehabilitation setting (meaning they reached the EFP), and (2) reporting this experience at conferences or in research papers.

All participants were informed about the aim of the study and that all participation was voluntary. All participants provided written informed consent before contributing to the study. Ethics approval was provided by the Sendai Seiyo College (No. 0211).

### 2.3. Procedure

Nine occupational therapists from the home rehabilitation unit were invited and interviewed about their experiences with one client from the start of home occupational therapy to achievement of their goals.

The occupational therapists from the home rehabilitation unit were asked to select a client for whom they had set and achieved an activity/participation-level goal within the past year. Subsequently, the participants were interviewed about their experiences from the start of the home-based occupational therapy to achieving their goals.

Individual interviews were conducted between August and December 2020 to understand the participants’ perspectives on goal-setting. Interviews lasted 30–60 min with each participant. The interviews were conducted by the first author (with experience in qualitative research as an occupational therapist and a professor of college, with 22 years of clinical experience), based on the interview guide (Table 2). The interview guide was designed to clarify the participants’ clinical decision-making regarding the process from the start of occupational therapy to goal setting and attainment, which is essential for EFP in this study.

Before the interview, the participants were asked to check their assessment results and progress based on the medical records of the client they chose. Interviews were conducted using the Zoom video conferencing system (Zoom Video Communications, Inc. San Jose, CA, USA) to prevent the spread of COVID-19. The data were recorded on a digital voice recorder and transcribed verbatim.

### 2.4. Data Analysis

The authors (YS, KT), with experience in qualitative occupational therapy goal-setting research, perused the transcripts. They were labeled with a focus on “Occupational therapists’ experience in relation to EFPs.” The labels were sorted chronologically, and a TEM graph was created for each of the nine participants using the TEM conceptual tool. The prototype TEM graphs were presented to the participants in the second and subsequent interviews for confirmation.

The EFP path was revised and analyzed repeatedly until “transview” saturation was reached, which was verified with the occupational therapist for any errors or inadequacies in the researcher’s interpretation and items that should be added. Transview is a state in which the researcher and participant integrate each occupational therapist’s perceptions through multiple discussions. No gap existed between the TEM graph created by the researcher and the participant’s perceptions. This is synonymous with “theoretical saturation,” used in representative qualitative studies. Furthermore, the TEM graph was completed after an average of three interviews with each participant.

The nine TEM graphs for each participant were integrated according to the method described in a previous study [10]. Similar content was integrated by increasing the level of abstraction of the labels. The paths experienced by all participants were designated as OPP.

## 3. Results

### 3.1. Participants’ Characteristics and Home Rehabilitation

The participants’ average clinical experience was 8.4 years (±4.2). The number of years of experience in home rehabilitation was 1.8 years (±1.9). The frequency of home visits was once or twice a week for all nine participants; however, the length of time before setting goals for activity, participation levels, and goal attainment varied (Table 3).

### 3.2. Definition of Items Using TEM Diagram

Table 3 presents the item tools in the TEM diagram and their definitions in this study. All items (EFP, polarized equifinality point, BFP, OPP, SD, and SG) were labeled by the authors based on the purpose of this study.

### 3.3. Classification of Time Phases

The pathways that describe the experiences of the nine occupational therapists are presented in Figure 2. The pathway processes were categorized into the following two phases: “goal setting” and “goal attainment.”

#### 3.3.1. Phase 1: Goal-Setting Process

Two of the nine occupational therapists had completed goal setting for in-home rehabilitation clients during their hospitalization. In contrast, the other seven occupational therapists began discussing the client’s wishes and challenges in daily living at the first visit.

Although some clients expressed their expectations and challenges, others stated that they did not want to do anything or only wanted functional training. Thus, BFP1 was set up as a situation in which it was difficult to focus on the activity and participation-level goal (Table 4). At this time, the occupational therapist carefully explained the purpose of occupational therapy and the significance of increasing activity and participation, considering the client’s level of understanding and psychological state. The occupational therapist assessed the client’s level of understanding, provided another opportunity for interviews and discussions, and set targets for activity and participation levels. Interestingly, all occupational therapists provided the opportunity for interviews. The occupational therapists believed it was important to listen to the clients, and for the clients to reflect on themselves and participate in the goal-setting process through the interview. The guiding principles for the occupational therapists’ behavior in Phase 1 are as follows:Paternalistic explanations and persuasion are not effective.Building a trusting relationship with the client is a priority.Client-centered values are the basis for interaction.“The goal of occupational therapy is not for me to decide.”“Occupational therapy is more effective when the client participates actively.”“It is important for the occupational therapist to provide the client with opportunities to reflect on the current situation and to work together to identify challenges and expectations related to their level of activity and participation.”

#### 3.3.2. Phase 2: Goal Attainment

In the first phase, there were various pathways; however, the pathway in Phase 2 was similar among all occupational therapists. The advantage of home occupational therapy is the ability to execute work directly in a client’s home. Although all occupational therapists appeared to prioritize actual movement, there were many difficulties (Table 4). Some clients were at risk of falling. Some clients shared the goal of activity and participation levels but wanted functional training. Some clients had severe functional impairments, which made interventions in actual activities difficult. Additionally, they were not always highly motivated. The occupational therapists considered various situations, made holistic judgments, and combined different intervention models, such as functional training, elemental movement practice, actual movement practice, and environmental adjustments, to achieve the goal.
ijerph-20-05746-t004_Table 4Table 4A part of the narrative in each BFP (Bifurcation Point).BFPSD and SG Related to BFPA Part of the Narrative Obtained from the ResultsBFP1: Difficulty in having discussions focused on activities and participation• SD1: CL is inactive with no hope• SD2: CL wants only functional training• I did not think it would be effective to forcefully persuade clients to focus on the activity and participation level goal.• I thought that building a trusting relationship first was a higher priority.• In this situation, if communication and intervention are focused on activity and participation level, it would only lead to client distrust.• I thought creating an environment for clients to reflect and resolve issues together was important.BFP2: Difficulty in having occupation-based interventions• SG3: Agree to an occupation-based intervention• SD5: High risk of falling• SD6: Physical dysfunction• SD7: Interest in functional recovery• I do not think the client’s reductionist approach can be changed by persuasion.• I think priority should be given to interventions to reduce the client’s risk of falling.• I think it is better to provide functional training and elemental movement exercises concurrently so that the client can participate in occupational therapy with conviction.BFP: bifurcation point, SG: social guidance, SD: social direction.


The occupational therapists who shared their goals for activity and participation levels before starting home-based rehabilitation did not face BFP1, as indicated in Phase 1. Instead, they spent time reviewing their goals with their clients at each visit. They also provided opportunities for the client’s family to actively observe the occupational therapy setting. Thus, the goals were shared before the start of the rehabilitation visit, and care was taken to ensure the client remained goal-oriented at all times. Similar to the other seven occupational therapists, they flexibly combined different intervention models, considering the client’s condition. The beliefs and values that served as the guiding principles for the occupational therapist’s behavior in Phase 2 are as follows:“The priority is an intervention aimed at risk reduction.”“When I visit a client’s home, I am the only staff member on-site; therefore, I need to consider various factors in determining interventions, including risk management, in addition to meeting activity and participation-level goals.”“Functional restoration training should be conducted concurrently to convince the client to participate in occupational therapy.”“The priority is to establish a trusting relationship. It was ineffective to forcefully persuade the client to focus on activity- and participation-level goals.”“Forcing communication and interventions to focus on activity and participation level would only lead to client distrust.”“I thought it was important to make a chance for clients to reflect and to resolve issues together.”Functional training and direct interventions at the activity and participation level are necessary to improve and maintain occupational performance.Occupational procedures and methods should be modified to make them safe and efficient.

### 3.4. Typification of Trajectories

We categorized the pathway into the following four types: (1) occupational therapists shared goals with the client before starting therapy, (2) occupational therapists could share the goal after the first interview, (3) occupational therapists could not share the goal, but they did after explanation and discussion with the client, and (4) occupational therapists could not set an activity- and participation-level goal (client expectations of recovering their physical function).

#### 3.4.1. Type I: Pathways with Shared Activity and Participation Goals before the Start of Home-Based Rehabilitation (Participants F and I)

This pathway was characterized by setting activities and participation goals before the start of home-based rehabilitation. During the hospital stay, the occupational therapists explained the purpose of home-based rehabilitation (OPP1). Additionally, the clients discussed the activity’s goals and participation with the occupational therapist. At the start of home rehabilitation, the intervention was focused on real occupational practice (OPP2); however, it also included functional training, elemental movement exercises, and environmental adjustments.

The home-based rehabilitation program was terminated after the goals were achieved (EFP). The occupational therapist on this pathway shared the activity- and participation-level goals with the client before beginning the home visit, but also spent time with the client on each visit, reviewing the occupational therapy goals with the client. They also encouraged family members to actively observe clients participating in occupational therapy to help them achieve their goals. The goal was clearly defined at the end of home-based rehabilitation.

#### 3.4.2. Type II: Pathways with Shared Goals for Activity and Participation Levels in Initial Interview Evaluations (Participants A, C, and G)

This pathway was characterized by goal setting after the start of home-based rehabilitation. At the start of home-based rehabilitation, the occupational therapist confirmed the client’s desires in daily life, and the goal-setting interview could begin smoothly.

During the interview assessment, the occupational therapists were cautious and superficially shared activity- and participation-level goals. During the interview (OPP1), the occupational therapist and client had difficulty selecting goals because many expectations were expressed (SD4). The occupational therapist believed handling this issue well rather than intervening immediately to clarify the occupational therapy program alongside encouraging the clients’ goal-oriented participation was important.

Occupational therapists could share goals in activity and participation levels through careful interviews (SG2). After sharing goals, the clients agreed to the interventions (SG3); however, the clients were at risk of falls (SD5), loss of physical function (SD6), and excessive expectations for functional training (SD7). The occupational therapists believed it was important to explain the risks to clients and have them perform the tasks with understanding. Subsequently, occupational therapy was implemented while combining functional training, elemental movement training, actual movement training, and environmental adjustment based on the client’s situation.

#### 3.4.3. Type III: Pathway with Activity and Participation Levels Goal Setting by Fully Explaining the Purpose of Home-Based Rehabilitation (Participants B and H)

This pathway was characterized by the client’s inability to discuss their activity- and participation-level goals at the beginning of home-based rehabilitation (BFP1). This was because even when occupational therapists discussed their expectations for daily living, the clients stated that they had no expectations (SD1) and wanted functional training (SD2). At this time, the occupational therapist considered that the client did not know the purpose of occupational therapy. Therefore, the occupational therapist explained this considering the client’s psychological state, being careful not to persuade them in a biased manner. Furthermore, even if the goal of the activity and participation level could be shared, they believed it was important to intervene to ensure that the client’s motivation was maintained and they would not experience failure.

During the goal-setting interview (OPP1), it was difficult for occupational therapists and clients to agree on their goals because of differences in expectations (SD4). However, occupational therapists could share the activity- and participation-level goals (SG2) through careful interviews. After sharing goals, as in other pathways, functional training, elemental movement training, actual movement training, and environmental adjustments were conducted based on the client’s situation.

#### 3.4.4. Type IV: Pathways with Clients Who Did Not Understand the Purpose of Occupational Therapy Even after Explaining in Interviews (Participants D and E)

This pathway was characterized by difficulty in understanding occupational therapy. Although occupational therapy confirmed clients’ expectations in daily life, some clients refused interventions other than functional training (SD2). The occupational therapist carefully explained the purpose of the intervention; however, the client refused to work with them (SD3). At this time, the occupational therapist considered forcing the discussion to focus on activity and participation levels to be a barrier to building partnerships. Thus, instead of explaining the importance of the activity and participation level and attempting to persuade the client, providing the functional training desired by the client first was important. While providing functional training, the occupational therapist conversed with the client to increase their interest in important activities and participation. The purpose of occupational therapy was explained by increasing the client’s intention to work. The client agreed to discuss their activity- and participation-level goals (SG1). Subsequently, the goals for the activity and participation levels were shared through interviews. Next, as with other pathways, interventions were implemented that included interwoven functional, elemental movement, and actual movement training to achieve the goal.

## 4. Discussion

In this study, we used TEM to visualize and analyze the trajectories of nine occupational therapists who set activity and participation level goals with their clients and helped them achieve them. We identified two phases and four pathways in setting and achieving activity and participation goals. Occupational therapists respected clients’ expectations, explained the purpose of occupational therapy in detail, and selected interventions flexibly when they faced barriers, such as difficulties in discussing activity and participation level goals or in implementing occupation-based interventions. This was discussed considering the two research questions described above.

### 4.1. Achieving Activity- and Participation-Level Goals

Our findings suggest that the goal-setting process is categorized into two phases (“Goal-setting process” and “Goal attainment”) and two BFPs (“Difficulty of discussing activity- and participation-level goals” and “Difficulty of conducting occupation-based intervention”). In the “Goal-setting process” phase, occupational therapists faced difficulties discussing activity- and participation-level goals. When they faced this hurdle, all nine occupational therapists set up a time for individual face-to-face interviews with their clients for goal setting, and explained the purpose of occupational therapy. All of the occupational therapists believed that interviewing goal setting was essential, even if they already knew about the client’s daily life and goals (type I). Since the expectations for activity and participation level varied for each client, occupational therapists felt it was important for clients to be involved in the goal-setting process.

Regarding the importance of involving clients in the goal-setting process, Palmadottir revealed that the positive outcomes of occupational therapy are experienced at various levels, and can be further augmented by optimizing the organization of the therapeutic process, establishing the purpose and meaning of the work, and fostering a strong client-therapist relationship. The study concludes that occupational therapists should prioritize clients’ work-related problems and needs and engage them more actively in a goal-oriented therapeutic process [11]. Rebeiro further emphasizes that centering solely on the illness rather than the person undermines the collaborative partnership between the client and healthcare professional and excludes the client from the decision-making process [12]. Therefore, we believe that the emphasis placed by the nine occupational therapists on focusing on activity and participation during the initial interview assessment was crucial for promoting collaboration between the therapists and clients toward enhancing activity and participation.

In the “Goal attainment” phase, occupational therapists faced the difficulty of conducting an occupation-based intervention. Home rehabilitation was performed in the client’s home. Therefore, direct interventions for occupational performance were possible. However, the clients had different situations, such as a strong desire for functional training, a high risk of falling, or the effects of disuse syndrome. Under these circumstances, the nine occupational therapists intervened directly in occupational performance and flexibly selected programs, including functional training, elemental movement training, and environmental adjustments.

In a systematic review conducted by Chang et al. on the efficacy of interventions provided within the scope of occupational therapy to enhance activities of daily living (ADL) performance in older adults residing in the community, individual OT interventions were found to include assessment, education and information, prevention strategies, exercise, assistive technology, home hazard modification, recommendations for assistive devices and services, coaching, and/or follow-up sessions. The study further observed that occupational therapists’ interventions were marked by their client-centeredness, empowerment, education/information dissemination, engagement in meaningful activities, and collaboration [13]. Chisholm et al. also found that in providing work-based interventions, it is important to use physical approaches to restore function. Supporting clients in performing ADL by adapting and modifying the environment, practicing vocational activities incorporating physical approaches to restore functionsupporting clients regarding cognitive, emotional, and behavioral aspects, supporting clients in self-management skills (e.g., nutrition, exercise, and stress management), promoting healthy lifestyles (e.g., healthy eating, exercise habits, limiting smoking, and alcohol consumption), and supporting clients using technology (e.g., computer-assisted therapy and the use of virtual reality) are important methods [14]. As supported by these results, it is necessary to provide multifaceted interventions, including those implemented by the nine occupational therapists for their clients.

### 4.2. Categories of Assessment and Intervention Processes for Occupational Therapists to Achieve Activity- and Participation-Level Goals in Home Rehabilitation

In the Type I pathway, clients were provided with an explanation of the purpose of home-based rehabilitation, and specific goals were shared during hospitalization. No apparent SD was found during the home-based rehabilitation process of this type, and the goals were achieved smoothly. As suggested by previous research, setting rehabilitation goals earlier can shorten the time taken to achieve these goals and facilitate their attainment [15,16,17]. From the results of these previous studies, it can be understood that the current statement is plausible, suggesting that the Type I pathway, in which goals are set before the start of home rehabilitation, is less problematic, and goal achievement is smoother. The Japanese government has noted the irregular continuation of functional training in home rehabilitation [2]. Targeting collaborative activities and participation levels between occupational therapists and clients at an early stage, as outlined in the Type 1 pathway, can address the issue of aimless and disorganized provision of home-based rehabilitation.

The Type II pathway smoothly set activity- and participation-level goals through the interview assessment; however, it did so to a lesser extent than Types III and IV. A survey from the Japanese government [18] revealed that the client’s issues varied widely from physical function to participation level. However, 60–70% of the home rehabilitation plans prioritized gait or maintaining posture, and <5% were activity and participation [18]. Clients frequently do not understand occupational therapy’s purpose or accept anything other than functional training, as in Types III and IV. Many clients want occupational therapists to provide functional training.

Although functional training may be necessary depending on the client’s condition, simply incorporating the participant’s expectations, which focus on physical function, will limit the possibilities of the amount of activity and level of participation. A systematic review conducted by Arbesmanr et al. [19] suggests that skill-based training is more effective when combined with occupational- and activity-based health management programs.

Occupational therapists should work with clients to provide sufficient information about the problems to be solved by occupational therapy and to set goals for activity and participation levels. However, some clients, including those in the Type IV pathway, may not understand the importance of focusing on the activity and participation level simply by carefully explaining occupational therapy. In such cases, occupational therapists should respect clients’ expectations, build rapport, and provide flexible occupational experiences to avoid conflict with clients. Based on these processes, clients can reflect on their occupations.

### 4.3. Limitation and Future Research

We applied qualitative analysis (TEM) using linguistic data from occupational therapists who set and achieved activity- and participation-level goals with their clients. The interviewer and participants developed a TEM diagram through interviews in this method. The TEM diagrams were assessed repeatedly during at least three interviews until transview saturation was reached, which may have biased the results. Additionally, the results cannot be generalized because this was a qualitative study with nine participants.

This study focused on the perceptions of occupational therapists engaged in home-based rehabilitation. In the future, it would be necessary to analyze the client’s experience in achieving the activity- and participation-level goal. In this study, some clients continued to use home-based rehabilitation even after achieving the goal. However, the issue of continuing functional training remains. Therefore, it is necessary to conduct a quantitative study to explore the factors for setting and achieving activity and participation goals to solve these issues.

## 5. Conclusions

This study categorized the goal-setting process in home rehabilitation into the following two phases: the “goal-setting process” and “goal achievement,” with two key barriers to progress identified as “difficulty discussing activity/participation level goals” and “difficulty providing occupation-based interventions.” In response to these challenges, occupational therapists utilized several approaches. During the goal-setting process, therapists respected clients’ expectations, provided detailed explanations of the occupational therapy process, and conducted individualized face-to-face interviews when necessary. For occupation-based interventions, therapists flexibly utilized various means, including functional training, elemental movement training, and environmental modification, to achieve the desired outcomes. Furthermore, the study identified four pathways to goal achievement and highlighted key points and processes for goal setting and achievement. The findings have practical implications for achieving activity- and participation-level goals in the home rehabilitation setting. Overall, this study has elucidated the goal-setting process in home rehabilitation, and the barriers that can impede progress. The strategies occupational therapists employ to overcome these obstacles and the pathways to goal achievement identified in this study can inform the development of effective interventions in the future.

## Figures and Tables

**Figure 1 ijerph-20-05746-f001:**
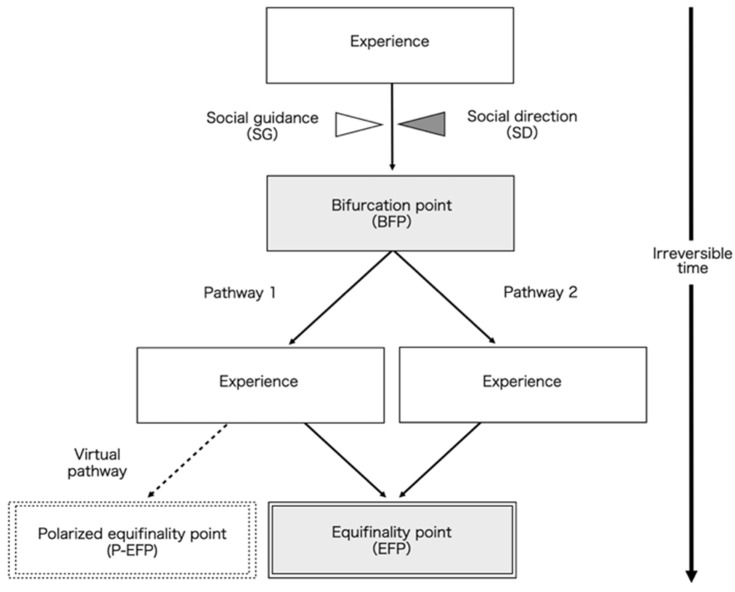
Example of trajectory equifinality modeling diagram.

**Figure 2 ijerph-20-05746-f002:**
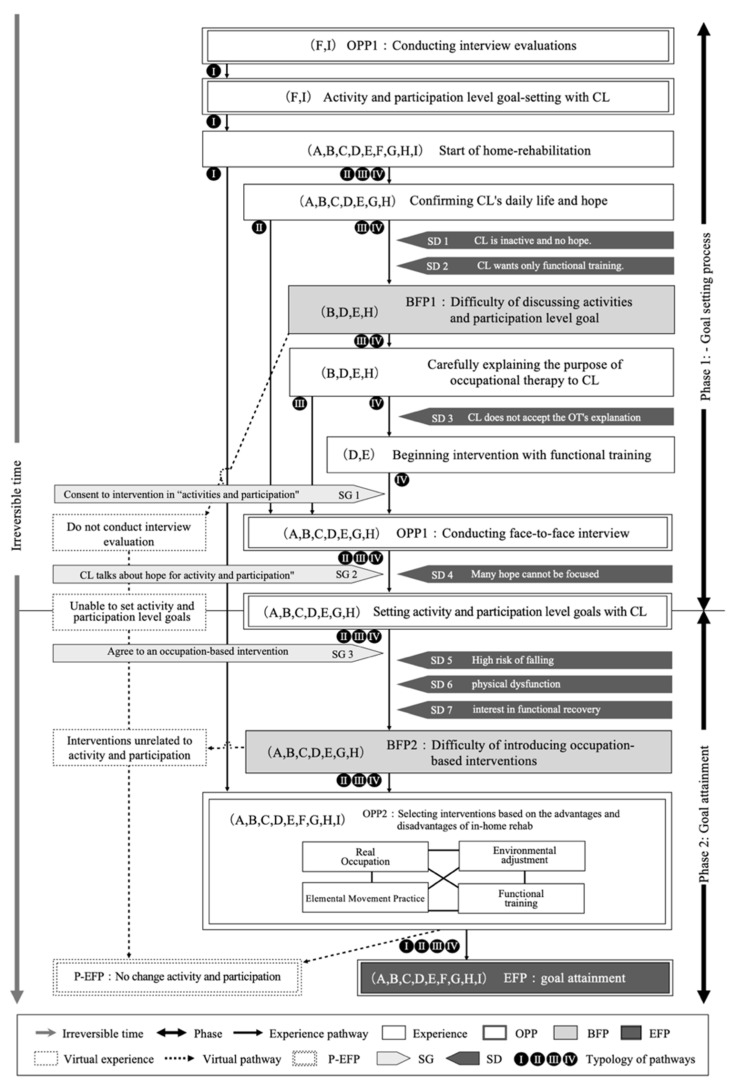
TEM diagram. OPP: Obligatory Passage Point, BFP: Bifurcation Point, EFP: Equifinality Point, P-EFP: Polarized Equifinality Point, SG: Social Guidance, SD: Social Direction.

**Table 1 ijerph-20-05746-t001:** Description of the TEM conceptual tool and its implications for this study.

Major Concepts	Definition	Implications for This Study
Irreversible Time	TEM incorporates the concept of time, but as an ongoing, qualitatively irreversible state rather than a concrete length of time reduced to units	−
EFP (Equifinality Point)	The situation that the study participants are in	Goal attainment
P-EFP (Polarized Equifinality Point)	The situation at the polar opposite of the equifinality point	No change in activity and participation
BFP (Bifurcation Point)	A situation at a certain experience that allows a choice between several optional trajectories	BFP-1: Difficulty in having discussions focused on activities and participation BFP-2: Difficulty in performing occupation-based interventions
OPP (Obligatory Passage Point)	A point that many people necessarily pass through	OPP-1: Conducting interview evaluations OPP-2: Selecting interventions based on the advantages and disadvantages of home-visit rehabilitation
SG (Social Guidance)	Environmental factors that attempt to move the individuals toward the equifinality point and their underlying social pressures	Environmental influences that are conducive to goal setting and goal attainment
SD (Social Direction)	Environmental factors that attempt to move the individuals away from the equifinality point and their underlying social pressures	Influences from the environment that act as inhibitors to goal setting and goal attainment

**Table 2 ijerph-20-05746-t002:** Interview guide.

Main Questions	Subtopics
1. What experience pathway did you follow before setting the activity- and participation-level goals?	− What is the goal? − How did you determine the goals? − What were some of the barriers to goal setting? − What did you do to overcome the barriers? − What kind of clinical judgment is the device based on? − What changes have resulted from this device?
2. What experience pathway did you follow after setting a goal for activity and participation levels until achieving that goal?	− What barriers have you encountered in the process of achieving the goal? − What did you do to overcome the barriers? − What kind of clinical judgment is the device based on? − What changes have resulted from this device?

**Table 3 ijerph-20-05746-t003:** Basic demographics of participants and overview of home rehabilitation.

Information		A	B	C	D	E	F	G	H	I
**Occupational therapist**	Number of interviews (times)	3	3	4	3	2	3	3	3	3
	Total interview time	110 min	190 min	150 min	120 min	90 min	150 min	110 min	110 min	120 min
	Sex	male	female	female	male	female	male	female	male	male
	Age (years)	38	26	29	37	29	27	28	32	36
	Clinical experience in OT (years)	14	4	7	14	4	5	6	9	13
	Experience in home rehab	4 years	6 months	1 month	1 month	2 years	1 year	1 month	3 years	5 years
	Pathway in TEM	Type II	Type III	Type II	Type IV	Type IV	Type I	Type II	Type III	Type I
**Client**	Diagnosis	Spiral cord injury	Stroke	Cervical spondylosis	Rheumatoid arthritis	Stroke	GBS	Stroke	Lumbar spondylosis	Femoral fracture
	Level of basic ADL	Partial assistance	Partial assistance	Partial assistance	Partial assistance	Partial assistance	Partial assistance	Independent	Independent	Independent
	Goal	Gardening	Work	Wash hair	Cooking	Meet to friends	Take a shower	Mobilize outdoors	Mobilize outdoors	Cooking
	Frequency of home rehab	2/week	1/week	2/week	1/week	1/week	3/week	1/week	1/week	2/week
	When is set a goal from starting OT?	After 1 week	At the start	After 2 weeks	After 2 months	After 5 months	At the start	After 2 weeks	After 5 months	After 6 months
	When did the goal achirve?	After 1 month	After 8 months	After 3 months	After 5 months	After 3.5 years	After 1 year	After 6 month	After 2 months	After 6 months
	After goal achievement	Continue	End	Continue	Continue	Continue	End	End	Continue	End

OT: occupational therapist, Rehab: rehabilitation, TEM: trajectory equifinality model, ADL: activities of daily living, GBS; Guillain–Barre syndrome.

## Data Availability

The data used to support the findings of this study are available from the corresponding author.

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
