# Peer review of "Longitudinal Process of Setting and Achieving Activity- and Participation-Level Goals in Home Rehabilitation in Japan: A Qualitative Study Using Trajectory Equifinality Modeling"

_ijerph, 2023, doi:10.3390/ijerph20095746_

Round 1

Reviewer 1 Report

The authors of the paper presented a qualitative study on setting and achieving goals in home-based rehabilitation. The study aimed to clarify the process of assessment and intervention and to stress its role in designing patients’ treatment goals.

In the manuscript, after a sound introduction authors presented the study design. The research was conducted using Trajectory Equifinality Model (TEM). It was also based on the 79 Consolidated Criteria for Reporting Qualitative Research checklist. 

The article is original and presents an interesting approach to rehabilitation,  stressing the role of designing patients’ treatment goals on the activity and participation level. The structure of the paper is correct. The language is comprehensive (only some minor spelling and stylistic errors appeared). The methods and results are clearly described. The limitations of the study were addressed in the article. 

The weakest point of the article however is the discussion. In each paragraph only one reference was used. It would be advised to update this part of the paper to show the obtained results in a broader perspective. I believe that the list of articles that are addressed in the discussion should be expanded. 

As authors stated in their work, this study is a qualitative research on a small group of subjects. The interviews were concentrated on the occupational therapists experience in rehabilitation goals setting and it is necessary to analyze also the patients’ opinions. The scientific soundness of obtained results is low, however it seems that presented study can be a base for designing a more complex, quantitive research on a larger group. 

Author Response

Comment

The authors of the paper presented a qualitative study on setting and achieving goals in home-based rehabilitation. The study aimed to clarify the process of assessment and intervention and to stress its role in designing patients’ treatment goals.

In the manuscript, after a sound introduction authors presented the study design. The research was conducted using Trajectory Equifinality Model (TEM). It was also based on the 79 Consolidated Criteria for Reporting Qualitative Research checklist.

The article is original and presents an interesting approach to rehabilitation, stressing the role of designing patients’ treatment goals on the activity and participation level. The structure of the paper is correct. The language is comprehensive (only some minor spelling and stylistic errors appeared). The methods and results are clearly described. The limitations of the study were addressed in the article.

Respond to reviewer-1

We thank you for your effort in reviewing our manuscript. We are pleased to hear that you found our study to be original and interesting. Regarding the minor spelling and stylistic errors that you mentioned, we have thoroughly reviewed the manuscript and made the necessary corrections. Your feedback has been invaluable in improving the quality of our paper, and we hope that the revised version meets your expectations.

Comment

The weakest point of the article however is the discussion. In each paragraph only one reference was used. It would be advised to update this part of the paper to show the obtained results in a broader perspective. I believe that the list of articles that are addressed in the discussion should be expanded.

Respond to reviewer-1

We appreciate your comment that a more in-depth discussion of the implications of our findings is needed. Accordingly, we have extended the discussion section to provide a more in-depth analysis of the clinical significance of our findings.

Comment

 As authors stated in their work, this study is a qualitative research on a small group of subjects. The interviews were concentrated on the occupational therapists experience in rehabilitation goals setting and it is necessary to analyze also the patients’ opinions. The scientific soundness of obtained results is low, however it seems that presented study can be a base for designing a more complex, quantitive research on a larger group.

Respond to reviewer-1

We thank you for your valuable comments regarding our manuscript. We appreciate your acknowledgment of the limitations of our study, particularly regarding its small sample size and qualitative nature. We agree that analyzing the patients' opinions on rehabilitation goal-setting in future research would be beneficial. We also appreciate your suggestion that our study could serve as a foundation for a larger quantitative study in the future.

Although our study may have its limitations, we believe that it still provides valuable insights into the role of goal setting in rehabilitation. We hope that our work can contribute to developing more effective rehabilitation practices in the future. Again, we thank you for your effort in reviewing our manuscript.

Reviewer 2 Report

In this interesting paper authors aimed to clarify the longitudinal goal negotiation and collaboration process to achieve activity- and participation-level goals. They conducted a qualitative study using the Trajectory Equifinality Model.

Comments:

Line 63: The ethical explanation is not clear here. Is there any ethical approval?

Results: tables need an explanation of all abbreviations.

Author Response

Comment

Line 63: The ethical explanation is not clear here. Is there any ethical approval?

Results: tables need an explanation of all abbreviations.

Respond to reviewer-2

We thank you for your feedback on our work. We appreciate your valuable comment about the ethical explanation and the tables in the results section.

Regarding the ethical explanation, we apologize for any ambiguity caused by the lack of clarity in that section. We obtained ethical approval for this study and have included a more detailed explanation in the manuscript to provide transparency and ensure ethical guidelines were followed.

Regarding the tables in the results section, we have revised them to include a clear explanation of all abbreviations used. This made it easier for readers to understand and interpret the results presented in the tables.

Again, we thank you for your feedback, and please let us know if you have any further suggestions or comments.

Reviewer 3 Report

Thanks to the team of authors for such interesting study on the challenging topic of setting and achieving activity and participation based goals in home rehabilitation. Acknowledging the novelty and significance of the topic, I have a few comments regrading the manuscript. 

1. The interesting study design is chosen based on Trajectory Equifinality Model (TEM) which is not widely known, in my opinion. Therefore I would suggest to reconsider the explanation and representation of this model in the manuscript text, because it is quite difficult to grasp all the abreviations used in the text, if the reader is not well familiar with this model (Table 3 is very helpful). 

2. I would like to suggest to reconsider the representation of the results in the Table 2 as it it very complex and difficult to perceive. Perhaps, attempt to reduce the text in the table (example- "Total interview time (min)" and only numbers in the cells; "Experience in the home rehabilitation (month)" and only numbers in the cell) as well as to divide some of the presented categories (example- to separate broad category in several parts: independence in P-ADL, Mobility indoor, Mobility outdoor, Use of AD), etc.

3. Conclusions almost shortly repeat the main results and sounds very general. 

Author Response

Comment

The interesting study design is chosen based on Trajectory Equifinality Model (TEM) which is not widely known, in my opinion. Therefore I would suggest to reconsider the explanation and representation of this model in the manuscript text, because it is quite difficult to grasp all the abbreviations used in the text, if the reader is not well familiar with this model (Table 3 is very helpful).

I would like to suggest to reconsider the representation of the results in the Table 2 as it it very complex and difficult to perceive. Perhaps, attempt to reduce the text in the table (example- “Total interview time (min)” and only numbers in the cells; “Experience in the home rehabilitation (month)” and only numbers in the cell) as well as to divide some of the presented categories (example- to separate broad category in several parts: independence in P-ADL, Mobility indoor, Mobility outdoor, Use of AD), etc.

Respond to reviewer-3

We thank you for your effort in reviewing our manuscript and providing this valuable feedback. We appreciate your suggestion regarding the presentation of the results in Table 2. We agree that the table may be complex and difficult to perceive, and we understand your point about reducing the text in the table and dividing some of the presented categories.

To address your concerns, we have considered your suggestions and revised the text to improve the representation of the results in Table 2. We have explored the possibility of reducing the text in the table and separating broad categories into several parts to improve readability and make it easier for readers to understand the results.

Again, we appreciate your feedback.

Comment

Conclusions almost shortly repeat the main results and sounds very general.

Respond to reviewer-3

We thank you for your insightful feedback on our manuscript. We appreciate your suggestion that the conclusions may repeat the main results and sound too general. Therefore, we have revised the conclusions by making them more specific and relevant to the research question and the study's overall purpose. Again, we thank you for your input, and please let us know if you have any further suggestions or comments.

Reviewer 4 Report

Thank you very much for allowing me to read this interesting manuscriptThe manuscript is based on a topic of great interest due to importance of Longitudinal Process of Setting and Achieving Activity- and  Participation-Level Goals in Home Rehabilitation. Congratulations for a comprehensive and well-written manuscript with a strong discussion. Nevertheless, there are some comments and questions that I would like to make:  

This model is associated with person-centred care, however, there is no mention of it during the introduction even though it is even referenced in the interviews.  

In the selection of the sample, it is observed that of the 9 participants, the experience in home rehabilitation  among them is very disparate, with three people who have only been working for a month and see persons once a week (two) or twice a week (one). How do the authors think this may affect speech of each OT and how has this been taken into account in the study, did they consider any specific inclusion or inclusion criteria? 

Tables with abbreviations should be included in the footnote.

Author Response

Comment

Congratulations for a comprehensive and well-written manuscript with a strong discussion. Nevertheless, there are some comments and questions that I would like to make:

This model is associated with person-centred care, however, there is no mention of it during the introduction even though it is even referenced in the interviews.

Respond to reviewer-4

We thank you for your constructive feedback on our paper. Accordingly, we have added a new reference [1] to the introduction and the following text:

The traditional approach to goal setting in healthcare relied on a biomedical model where clinicians made decisions independently, leading to conflicting expectations between patients and healthcare providers and frequently resulting in patients' goals being unfulfilled. However, the current trend is towards a patient-centered approach to goal setting, which prioritizes patients' needs and expectations. This collaborative process involves healthcare professionals, patients, and their families or caregivers as appropriate. Research has demonstrated that involving patients in goal formulation results in the development of more personalized goals and increased satisfaction with the rehabilitation experience [1].

Comment

In the selection of the sample, it is observed that of the 9 participants, the experience in home rehabilitation  among them is very disparate, with three people who have only been working for a month and see persons once a week (two) or twice a week (one). How do the authors think this may affect speech of each OT and how has this been taken into account in the study, did they consider any specific inclusion or inclusion criteria?

Respond to reviewer-4

We thank you for your valuable comment. "Our participants were selected based on the following two criteria: 1) having experience presenting at conferences on home rehabilitation, and 2) actively engaging in practices that focused on activities and participation, which is the specific aim of this study. Even if some participants had limited experience with home rehabilitation, they had ample experience as occupational therapists, making them suitable for the study. Additionally, the frequency mentioned in the study refers to the intervention time with the clients, not the working hours of the occupational therapists."

Comment

Tables with abbreviations should be included in the footnote.

Respond to reviewer-4

We thank you for your comment. Regarding the tables in the results section, we have revised them to include a clear explanation of all abbreviations used. This made it easier for readers to understand and interpret the results presented in the tables.

Again, we thank you again for your feedback, and please let us know if you have any further suggestions or comments.

Round 2

Reviewer 1 Report

Authors provided significant corrections to the manuscript. I believe that all of the reviewers' remarks were properly addressed. Expanding the discussion and additional explanations in the material and methods section certainly improved the quality of proposed manuscript. 

Reviewer 3 Report

Thanks to authors for revision of the manuscript! I found all the changes as sufficient.